# Does *Lactobacillus* Exert a Protective Effect on the Development of Cervical and Endometrial Cancer in Women?

**DOI:** 10.3390/cancers14194909

**Published:** 2022-10-07

**Authors:** Karolina Frąszczak, Bartłomiej Barczyński, Adrianna Kondracka

**Affiliations:** 1I Chair and Department of Oncological Gynaecology and Gynaecology, Medical University in Lublin, Staszica 16, 20-081 Lublin, Poland; 2Department of Obstetrics and Pathology of Pregnancy, Medical University in Lublin, Staszica 16, 20-081 Lublin, Poland

**Keywords:** cervical cancer, endometrial cancer, dysbiosis, *Lactobacillus*

## Abstract

**Simple Summary:**

Cervical cancer is the fourth most common cancer in women worldwide. Tumour-related deaths are most frequent in low- and middle-income countries. Currently, most vaccines against human papillomavirus (HPV) are based on virus-like particles; they protect against HPV infection but have no therapeutic effects. Because dysbiosis has been shown to increase cancer risk, lactic acid bacteria (LAB)-based vaccines, which have been shown to have an immunomodulatory effect, have recently attracted attention. Mucosal immunization with viable colonies of *Lactobacillus* via intranasal, intravaginal, and oral routes to decrease the risk of cervical cancer seems promising; thus, such research is of high value. While advances have been made in understanding associations between microbiota dysregulation and carcinogenesis, further studies are required to identify the underlying cellular mechanisms and to confirm previous findings. This manuscript summarizes available data concerning the impact of microbiota on cancer risk and presents recent strategies to fight cervical and endometrial cancers.

**Abstract:**

Cervical cancer is a significant health problem with increasing occurrence and mortality. This infection-associated tumour is caused by the human papillomavirus (HPV). HPV infection is cleared by the immune system within 6–18 months in most patients; however, persistent high-risk HPV (hrHPV) infections can lead to the development of cervical cancer. Virus persistence is promoted by immunodeficiency, *Chlamydia trachomatis* infection, smoking, and age, as well as the imbalance of cervicovaginal microbiota and inflammation. The abundance of bacteria in the vagina favours the maintenance of a dynamic balance; their coexistence influences health or disease states. The eubiotic vaginal microbiota of reproductive-aged women is composed mostly of various *Lactobacillus* species (spp.), which exert protective effects via the production of lactic acid, bacteriocins, polysaccharides, peptidoglycans, and hydrogen peroxide (H_2_O_2_), lowering pH, raising the viscosity of cervicovaginal mucus, and hampering both the adhesion of cells to epithelial tissue and the entry of HPV. The depletion of beneficial microorganisms could increase the risk of sexually transmitted infections. Emerging therapies involve mucosal, intranasal vaccines, which trigger systemic and mucosal immune responses, thus protecting against HPV-induced tumours. The use of probiotics has also been suggested to affect various biological processes associated with tumourigenesis (inflammation, oxidative stress, apoptosis, proliferation, and metastasis).

## 1. Introduction

Cervical cancer is an important health issue since according to statistics, it is the third most common cause of tumour-related deaths in females worldwide [1]. The estimated prevalence of this disease reaches 530,000 new cases, while approximately 275,000 females die from cervical cancer annually [2]. This infection-associated neoplasm is caused by strains of the human papillomavirus (HPV). Out of over 100 HPV strains identified, 13 (including HPV-16 and -18) were found to be the cause of cervical cancer in 100% of cases [3,4]. In most cases (90%), HPV infection appears to be cleared by the immune system within 6–18 months [5]. The launch of various host mechanisms, including the activation of toll-like receptors (TLRs) and natural killer (NK) cells, has been demonstrated to be sufficient in many cases for eradicating HPV [6]. The initial immunity to HPV infection is provided by a local innate immune system. The frequency of HPV infections can be decreased by HPV vaccinations, which boost acquired immunity. However, persistent high-risk HPV (hrHPV) infections can lead to the development of cervical cancer, either directly via altering the cellular structure or indirectly via chronic inflammation and an immune escape [7,8]. Virus persistence is promoted by immunodeficiency, *Chlamydia trachomatis* infection, smoking, and age. Imbalances in cervicovaginal microbiota and inflammation were also demonstrated to play a key role in the modulation of virus persistence and consequent cancer development [1,3]. Under physiological conditions, the abundance of bacteria in the vagina favour the maintenance of a dynamic balance. However, when the balance of the vaginal microecosystem is disturbed, it can lead to the development of various gynaecological diseases, including the inflammation of the vagina, high-grade cervical intraepithelial neoplasia, and cervical cancer [9]. Numerous studies confirmed the differences in vaginal flora, including the abundance of *Staphylococcus epidermidis*, *Mycoplasma genitalium*, *Mycoplasma hominis*, *Escherichia coli*, enterococci, and *Bacteroides* species in females with cervical cancer compared with healthy individuals [10,11]. The aim of this review was to summarize the existing and emerging data concerning the impact of microbiota, especially *Lactobacillus* and dysbiosis generally, on the risk of cervical and endometrial cancer development as well as novel experimental therapies. We searched PubMed, Medline, and Cochrane databases using the following keywords: cervical cancer, endometrial cancer, dysbiosis, *Lactobacillus*, HPV, herpes simplex, and vaginal microbiota.

## 2. Origins of the Microbiome

Numerous microorganisms inhabiting the human body, especially the intestinal compartment, have been proven to be beneficial for health [12]. Over 1000 distinct bacterial species reside in the digestive alone [13]. Due to the great variability of microbiome composition across individuals, there is no unified definition of “healthy microbiota” [14]. The settlement and development of gut microbiota (not only comprising primarily bacteria but also some fungi, viruses, and archaea) start during the first 3 years of life, eventually adopting an adult-like profile [15,16]. The dynamic, non-random process can be affected by different perinatal conditions, including the method of delivery, feeding method and diet, the use of antibiotics, mother’s age, metabolic status, lifestyle, and genetics [12]. The results of studies indicate that the first colonisation with the maternal microbiota starts at foetal age. Nagpal et al. [17] demonstrated that the abundance of lactobacilli in meconium was higher in vaginally delivered (VG) neonates compared to caesarean-section-delivered (CS) newborns. Infants who are delivered vaginally become exposed to the maternal vaginal and fecal microbiota, while those delivered via caesarean section come into contact with environmental microorganisms from maternal skin, hospital environment, and hospital staff, which is the reason for differences in neonatal gut colonization in both cases [18,19,20]. It was suggested that the presence of *Lactobacillus* and *Prevotella* microbiota in infants could be associated with the vertical transmission of species that inhabit the maternal vaginal tract [21]. Gut microbiota also differed between breastfed and formula-fed infants. “Milk-oriented microbiota” rich in *Bifidobacteria* was found in breastfed infants, while gut microbiota in the latter group was more diverse and contained staphylococci, enterococci, bacteroides, enterobacteria, clostridia, and the genus *Atopobium* [22,23,24]. Even after stabilization, gut microbiota may be affected by various factors, including the use of antibiotics, diet, age, stress, and some diseases, as well as environmental parameters including oxygen levels/redox state, pH, and temperature [12]. These microorganisms are vital for human health since they enhance the accessibility of certain nutrients, promote xenobiotic metabolism, prevent pathogen colonisation, and regulate and augment innate and adaptive immunological processes [12,13,25,26]. Early-life microbiota is vital for programming the immune system, intestinal tract development, and metabolism. The disruption of gut microbiota homeostasis in childhood has also been found to affect health state in adulthood, impairing the immune system and increasing the risk of metabolic disorders. Moreover, the persistent disturbance of the gut’s microbial community (known as dysbiosis) is associated with cardiovascular disease, inflammatory bowel disease (IBD), obesity, diabetes, cancer, and central nervous system disorders [13]. Therefore, strategies to alter maternal vaginal and fecal microbiota during pregnancy, including treatments with *Lactobacillus rhamnosus* during the second and third trimester of pregnancy, have been developed. Such therapy methods not only helped to maintain low vaginal pH and a pathogen-free vaginal environment but were also associated with *Bifidobacteria* colonisation in infants’ intestines [27,28,29].

## 3. The Microbial Environment of the Vagina and Upper Reproductive Tract

The female reproductive tract is inhabited by various coexisting microorganisms, which influence health or disease states [30]. The vaginal microbiota in healthy women of reproductive age is not diverse and usually comprises one or few *Lactobacillus* spp. [26,31]. During eubiosis, the vaginal microbiota of reproductive-aged women is primarily composed of various *Lactobacillus* spp., including *Lactobacillus gasseri, Lactobacillus crispatus, Lactobacillus jensenii,* and *Lactobacillus iners* [31,32,33]. Studies have shown that the depletion of beneficial microorganisms could be associated with a higher risk of sexually transmitted infections, pelvic inflammatory disease, preterm births, and spontaneous miscarriages [34]. According to studies, the profile of each female vaginal microbiome can be classified into six community state types (CSTs) [3,35]. *Lactobacillus*, especially *L. crispatus*, *L. gasseri*, *L iners*, and *L. jensenii,* is predominant in CST-I, II, and III, *Streptococcus* and *Prevotella* dominate in CST IV-A, and *Atopobium* is highly prevalent in CST IV-B. The presence of bacteria belonging to CST-IV is frequently associated with bacterial vaginosis. The aforementioned *Lactobacillus* species appear to be adapted for dominance in the vaginal niche since other types of *Lactobacillus* are not observed there [36,37]. The explanation of this phenomenon is unknown; however, it could be related to evolutionary issues [38]. The predominance of vaginal *Lactobacillus* spp. protects this microenvironment against the invasion of pathogens. It has been observed that *Gardnerella vaginalis* can also be dominant in the vaginal microbiome. The vaginal microbiome that is non-*Lactobacillus*-dominant appears to be more frequent in Hispanic and Black women (30–40%) compared to White and Asian women (10–20%) [39,40,41]. Ethnic and racial disparities can stem from different environmental and socioeconomic factors as well as diverse behaviour, e.g., sexual and hygiene-related [42]. However, some reports have indicated that at least one *Lactobacillus* can be related to disease states. For example, *L. iners* was identified in females with disorders of the vaginal environment [43,44,45]. The presence of *L. iners*-dominant vaginal microbiome is frequently observed during the transition to the non-*Lactobacillus*-dominant communities [46]. 

The vaginal microbiome can be affected by numerous factors, including infections with HPV and other STIs, sexual activity, lubricant use, the number of sexual partners, contraception use, hygiene practices, access to health care, diet and nutrition (fat-rich diet and high glycaemic load), smoking, physical activity, obesity, and alcohol consumption. Age; genetic and epigenetic factors; hormone levels; pregnancy; immune system impairment; stress; and exposure to xenobiotics, carcinogens, toxins, and antibiotics also influence its composition [47,48,49]. The vaginal microbiota profile depends on ethnicity; the *Lactobacillus* species are more prevalent in Caucasian and Asian women compared to Hispanic and Black women [3]. The ethnic differences in microbiota can be associated with either genetic factors affecting mucosal immunity and metabolic pathways or hygiene practices [3]. The gut microbiome has been demonstrated to indirectly influence the abundance of *Lactobacillus* in the vaginal microenvironment via the modulation of oestrogen release, which may imply the existence of a gut–vaginal axis [50,51,52]. β-glucuronidase and β-glucosidase secreted by microorganisms attach to oestrogen, thus leading to its enhanced reabsorption into the circulation [53,54]. In turn, unbound oestrogen reaches the female reproductive tract where it activates intracellular signalling associated with increased glycogen syntheses, thickening of the genital epithelium, and the production of mucus. Thus, females’ hormones, including oestrogen and progesterone, modulate vaginal colonisation with *Lactobacillus* spp. Higher levels of these hormones are associated with lower vaginal microbiota diversity and the dominance of *Lactobacillus* [55,56]. The relationship between oestrogen levels and the amount of vaginal *Lactobacillus* is mirrored by the finding of reduced *Lactobacillus* abundance in females before menstruation, i.e., when oestrogen levels are significantly reduced [57,58]. In this period of decreased oestrogen levels, some species become enriched, while others are depleted in the vaginal environment [35]. Temporal oestrogen deficiency may cause vaginal atrophy, which is partly responsible for higher bacterial diversity [35]. The decrease in the lactic acid bacteria pool is associated with the predominance of anaerobic bacteria and the subsequent risk of cervical cancer development. Though the mechanisms underlying hormone-related microbial composition of the vagina are not fully understood, it has been suggested that the dominance of *Lactobacillus* spp. may be associated with the oestrogen-driven maturation of vaginal epithelium, the production of α-amylase, and the accumulation of glycogen [59]. The degradation of glycogen by α-amylase to simple products such as maltose, maltotriose, maltotetraose, and α-dextrins promotes *Lactobacillus* growth and colony formation [60]. The use of synthetic hormones, e.g., contraceptives, has also been reported to decrease the incidence or recurrence of bacterial vaginosis [61]. In turn, smoking, sexual intercourse, and vaginal douching appear to diminish the abundance of *L. crispatus*, increase species diversity, and enhance the risk of bacterial vaginosis [62,63,64]. 

Data concerning common microbiota inhabiting the uterus, fallopian tubes, or ovaries are limited due to problems with its assessment [65]. The microbiota of the female upper reproductive tract was found to be very different from that of the vagina in composition and quantity [42]. Chen et al. [66] suggested that the number of bacteria in the uterus could be ~10,000-fold lower compared to the number of bacteria in the vagina. However, this estimation could be inexact due to the high risk of cross-contamination with bacteria from the lower part of the tract during transcervical collection. Moreover, it has been suggested that upper reproductive tract microbiota are more diverse compared to that of the lower tract; however, genuine members have not been identified since various studies indicated different microbiota compositions [66]. *Lactobacillus* species were also found in the upper tract, but their abundance gradually reduced with its withdrawal from the vagina and cervix. 

Numerous studies revealed that various body sites can serve as possible reservoirs of genital microorganisms. For example, common vaginal bacteria, including *Lactobacillus, Gardnerella, Sneathia, Prevotella, Atopobium, Gemella, Peptoniphilus,* and *Finegoldia*, are normally found in the urinary tract in both women and men [67,68,69]. Thomas-White et al. [69] observed that vaginal and bladder microbiota displayed comparable functional capacities, which differed from gut microbiota. The presence of *Lactobacillus* spp. in the bladder and the vagina could exert a protective effect against invading uropathogens. Moreover, the co-colonisation of both the vagina and rectum with vaginal *Lactobacillus* species, including *L. crispatus, L. jensenii, L. iners,* and *L. gasseri,* was associated with the lowest prevalence of bacterial vaginosis [70,71]. Therefore, it was suggested that the rectum might be a vital reservoir for vaginal lactobacilli. The presence of the vaginal microbiome’s members on male penile skin, in semen, and in urine specimens may imply that sexual partners can exchange microbiota residing in their urogenital tracts [72]. According to some studies, the composition of endometrial microbiota may affect implantation, pregnancy, and live birth rates [73]. *Lactobacillus*-dominated endometrial fluid and vaginal aspirate correlate with better outcomes. Uterine microbiota was suggested to exert an impact on the immune environment during conception [74]. Modifications of microbial composition in the endometrial fluid can elicit an inflammatory response within the endometrium, thus lowering the probability of embryo implantation success [75].

## 4. The Role of *Lactobacillus* in the Female Reproductive Tract

In contrast to many parts of the body in which great microbial diversity appears to be beneficial, in the vagina, a higher diversity of microbiota frequently results in dysbiosis and the development of disease states. Many studies have demonstrated that vaginal microbiota, including *Lactobacillus,* is involved in the protection of the reproductive tract and gastrointestinal tract against opportunistic infections [1,7]. The ability of *Lactobacillus* to produce lactic acid via the fermentation of glucose (glycolysis) supports vaginal eubiosis, as this organic acid helps preserve the vaginal acidic environment [76]. The acidic environment constrains the growth of some potentially pathogenic species, including *C. trachomatis, G. vaginalis,* and *Neisseria gonorrhoeae* [32,77,78,79]. Vaginal pH exceeding 5.0 was found to increase the risk of HPV in premenopausal women by 10–20% [80]. This finding could be partly explained by the fact that the HPV protein crucial for viral transformation, E5, is vulnerable to low pH [81]. Moreover, it offers optimal conditions for the metabolic functioning of cervical and vaginal cells [82]. Apart from affecting the pH of the environment, the chemical structure of lactic acid itself may modulate the HPV infection and the development of squamous intraepithelial lesions [3]. As a chiral molecule, lactic acid can be produced in the form of D- and L-isomers. Studies demonstrated that high levels of D-lactic acid could protect against *Chlamydia* infection and upper reproductive tract infections via the modulation of extracellular matrix metalloproteinase inducer (EMMPRIN) production in vaginal epithelial cells [83,84]. A higher L-lactate-to-D-lactate ratio is associated with the enhanced expression of EMMPRIN as well as the activation of matrix metalloproteinase 8 (MMP-8), eventually resulting in impaired cervical integrity and the easier entry of HPV into basal keratinocytes [83]. Nunn et al. [85] revealed that the predominance of *L. crispatus* and relatively high levels of D-lactic acid could increase the viscosity of cervicovaginal mucus, resulting in viral particle trapping. Lactic acid also limits the cytotoxicity of natural killer (NK) cells, diminishes the synthesis of pro-inflammatory cytokine IL-12, and promotes the release of anti-inflammatory interleukin-10 (IL-10) [86,87]. Apart from lactic acid, beneficial microbiota can also release other antimicrobial peptides, including bacteriocins and hydrogen peroxide (H_2_O_2_) [88,89]. Bacteriocins exert direct bactericidal effects, but they can also modulate the inflammatory immune response and mediate acquired immune response [1]. They possess anti-tumour properties resulting from cytotoxicity and the stimulation of cell lysis. Gassericin (bacteriocin), produced by *L. gasseri* as well as other strains of *L. crispatus* and *Lactobacillus reuteri,* acts on Gram-negative and Gram-positive bacteria [90,91]. Apart from bacteriocins, some bacteria (e.g., *Lactobacillus*) can also release biosurfactants, which modify surface tension, therefore hampering bacterial adhesion, biofilm formation, and the excessive growth of pathogenic anaerobes [92]. *Lactobacillus* epithelium adhesin (LEA), produced by *L. crispatus,* prevents the pilus-mediated adhesion of *G. vaginalis* [93]. The aforementioned bacteriocins and biosurfactants have also been demonstrated to disturb viral infiltration [94]. Moreover, both bacteriocin and surface-active components can constrain the synthesis of tumourigenic substances [95]. A higher rate of bacterial vaginosis was reported in females with decreased vaginal levels of bacteria capable of producing H_2_O_2_ [96]. The release of a variety of antimicrobial peptides (AMPs) into the uterine cavity poses a vital defence mechanism, protecting epithelial tissues against proteolytic enzymes secreted by pathogens [97,98]. Some studies have suggested that hypoxia could also promote the development of bacterial vaginosis since, in such conditions, bacteria are not able to produce H_2_O_2_ in a sufficient amount to inhibit pathogenic bacteria growth [99,100]. The interaction of commensal bacteria with endometrial epithelial cells was found to form an antimicrobial barrier against pathogens [101]. The presence of *Lactobacillus* in the vagina is associated with protection against the adherence of pathogenic bacteria to the epithelial tissue. These bacteria compete against pathogenic microorganisms for territories and nutrients [102]. *Lactobacillus* that occupies the vaginal epithelial cells (VECs) has been found to prevent the conglutination of invasive pathogenic bacteria, thus hampering the initiation of malignant tumours [103,104]. *Lactobacillus* was demonstrated to hinder the proliferation of malignant tumours via the secretion of phosphorylated polysaccharides, exopolysaccharides, and peptidoglycans [87,105]. Moreover, these bacteria can stimulate nitric oxide (NO) production by macrophages and impair energy metabolism in cancer cells [106]. Commensal bacteria stimulate the production of neutral, stable mucous by endometrial cells as well as preserve tight junctions [65,107]. An intact epithelial barrier is crucial for protection against the penetration and colonisation of opportunistic microorganisms. Furthermore, commensal bacteria can modify immune responses at the cellular level [101]. Studies have demonstrated that *Lactobacillus* enhances the proliferation and differentiation of thymus-derived cells (T cells) and ameliorates the immunological recognition and proliferation of B cells [108,109]. The adhesion of *Lactobacillus* and the absorption of nutrients have been demonstrated to trigger the complement system, which subsequently regulates microbial growth [110]. 

Motevaseli et al. [111] demonstrated that vaginal lactobacilli (*L. gasseri* and *L. crispatus*) could exert cytotoxic impact on cervical tumour cells, however, normal cells remained unaffected. Moreover, they observed that this effect was independent of lactic acid and pH. Studies have demonstrated the antimetastatic and antiproliferative properties of *Lactobacillus*, its subgenera, and its supernatants [87]. Via the modulation of HPV oncogenes, *Lactobacillus* was shown to limit cervical cancer cell viability. Another study has implied that *L. crispatus* is highly resistant to the co-colonisation of other bacteria and the transition into CST IV [46]. These bacteria are rarely found to coexist with other species. Furthermore, females with these bacteria have the lowest vaginal pH and are not susceptible to infections with bacterial STIs, HPV, herpes simplex virus-2 (HSV-2), or HIV [31,112]. Since bacterial vaginosis promotes the shedding of HIV and HSV-2, it has been suggested that dysbiosis and the reduced abundance of *Lactobacillus* may support the formation of an environment that induces the persistence of infections and leads to the development of squamous intraepithelial lesions [113]. The basic beneficial effects of Lactobacillus in the lower female genital tract are presented in Figure 1.

## 5. The Impact of Human Papillomavirus and Vaginal Microbiota on the Development of Cervical Cancer

The balance of vaginal microbiota is dynamic. Females are capable of recovering from lenient vaginal dysbacteriosis; however, if this state persists, it can stimulate the development of gynaecological cancer [9]. The diminished quantity and/or activity of *Lactobacillus* is associated with the overgrowth of anaerobic bacteria including *Atopobium vaginae, Gardnerella, Fusobacterium* spp., and *Sneathia,* as well as an enhanced risk of carcinogenesis [35,114,115]. Following colonisation, the anaerobic bacteria produce metabolites and enzymes that impair this barrier, thus enabling the entry of HPV. The preservation of the cervical epithelial barrier function hampers the entry of HPV into basal keratinocytes [116]. Dysbiosis has been demonstrated to support the development of HPV infection (HPV colonisation, clearance, persistence, and host immune response), thus increasing the risk of cervical cancer [42]. It appears that the combination of microbiome dysregulation, HPV infection, and the presence of inflammation is required to successfully trigger the development of cervical cancer. The presence of dysbiosis translates into altered microbial metabolites in the cervix and vagina as a result of the increased ratio of anaerobic to microaerobic bacteria [117]. Instead of lactic acid, new dominant bacteria produce amines [118,119]. Studies confirmed the importance of dysbiosis in the development of cervical cancer. One indicated that nearly three-fourths of females diagnosed with cervical cancer had disturbed vaginal microbiome [120]. Females with dysbiosis were reported to have higher levels of vaginal proinflammatory cytokines compared to those without [121]. The presence of chronic inflammation has been linked to carcinogenesis in various parts of the body [122]. Caselli et al. [123] demonstrated that patients with precancerous lesions and cervical cancer had increased levels of proinflammatory cytokines. Females with cervical intraepithelial neoplasia (CIN) showed elevated levels of IL-1α, IL-1β, IL-6, IL-8, and TNF-α in the vagina compared to healthy individuals. Nuclear factor kappa B (NF-κB) appears to be important during HPV infection. This virus was revealed to abolish the inhibitory effects of the immune system to freely replicate, promoting a state of persistent infection [124]. However, the transformation to high-grade intraepithelial neoplasia and cervical cancer requires NF-κB reactivation for the expression of genes involved in proliferation, VEGF-dependent angiogenesis, metastasis, and cell immortality [124]. Studies have indicated that some probiotics, such as *Bifidobacterium longum, Lactobacillus johnsonii, Lactobacillus plantarum, Lactobacillus fermentum,* and *Lactobacillus delbrueckii*, can inhibit various signalling pathways (including NF-κB), thus decreasing inflammation [125,126]. Apart from NF-κB, the signal transducer and activator of transcription 3 (STAT3) may also be involved in cervical cancer development, particularly in the transformation of precancerous cervical lesions into cancer [127,128,129].

Dysbiosis is also associated with enhanced oxidative stress, which results in the damage of DNA. Oxidative stress, together with proinflammatory cytokines, promotes the formation of a milieu appropriate for the onset or progression of cancer [130]. Dysbiosis has been found to impair the function and structure of key vaginal epithelial cytoskeletal proteins, thus facilitating HPV entry. Recent studies have provided evidence for a link between CSTs III and IV and the presence of HPV infection and subsequent development of preinvasive cervical disease [131,132,133]. The study of females with low- or high-grade squamous intraepithelial lesions and invasive cervical cancer (ICC) confirmed the role of microbial imbalance in disease development and progression. In women with cervical intraepithelial neoplasia, disease severity correlated with an increased diversity of vaginal microbiota and with the reduction in the relative amount of *Lactobacillus* spp. [133]. Moreover, the incidence of CST IV was higher in females with CIN and cervical cancer compared to healthy controls. Moreover, the study of the vaginal microbiota collected from premenopausal women and HPV-discordant twins demonstrated a markedly increased diversity of microorganisms, but lower amounts of *Lactobacillus* spp., in HPV-positive women compared to HPV-negative women [131]. Enrichment in *Fusobacteria*, including *Sneathia* spp., was suggested to be a probable microbiological marker related to HPV infection. Studies indicate that the abundance of *Sneathia* spp., belonging to *Fusobacterium* spp., is also associated with squamous intraepithelial lesions and cervical cancer [7,131,133]. These species are capable of producing a virulence factor, FadA, which modulates cell proliferation, migration, and survival via WNT signalling pathways [134]. Additionally, in a study of Mexican females with either squamous intraepithelial lesions or cervical cancer, greater diversity and increased relative levels of *Sneathia* spp. and *Fusobacterium* spp. were associated with higher disease severity [7]. A similar effect was observed in cases of high levels of *Anaerococcus tetradius, Sneathia sanguinegens,* and *Peptostreptococcus anaerobius*. The abundance of these bacteria was found in high-grade CIN [133]. Another study found that the incidence of CST IV doubled in low-grade squamous intraepithelial lesions (LSIL) and was even higher in highly squamous intraepithelial lesions (HSIL) and cases of invasive cancer [133]. The disturbance of cervicovaginal microflora supports the development of cervical cancer by altering vaginal acidity, the release of cytokines, immunosuppressive factors, local immunosuppression, and HPV persistence [45]. Numerous studies have indicated that the profile of the vaginal microbiota affects local immunity and can either prevent or promote HPV clearance and cervical cancer development [9,135]. Audirac-Chalifour et al. [7] demonstrated increased levels of IL-4 and TGF-1β mRNA in females possessing a greater relative abundance of *Fusobacterium* spp., which could enable HPV immune evasion and subsequent disease development. In turn, in Korean women with CIN, the substitution of *L. crispatus* with *L. iners, G. vaginalis,* and *Anaerococcus vaginae* was found to be a highest risk combination for the development of CIN [43].

Persistent HPV infection promotes the dysregulation of both cervical and vaginal microbiomes as well as mucosal metabolism, triggering a series of inflammation-related mechanisms including the pro-inflammatory cytokine-mediated activation of local mucosal immunity and the stimulation of overexpressed NK cells and macrophages [136]. However, such infections with HPV are not sufficient to cause cervical cancer; other contributing factors are also required, e.g., reduced abundance of *Lactobacillus,* enhanced production of substances that impair the structure and barrier function of cervical and vaginal epithelium mucosa, and pro-inflammatory cytokines that further disturb the epithelial intimal barrier [1]. A higher prevalence of cervical cancer may be associated with recurrent mixed microbial infections that can stimulate the replication, transcription, and modification of HPV [1]. CIN is facilitated by the presence of an inflammatory state and damage to epithelial cells [135,137]. Chronic inflammation resulting from persistent infection produces cytotoxic effects on normal cells and can damage DNA, leading to the initiation of tumourigenesis. Under the influence of epigenetic or microbial factors, autocrine and paracrine signals are triggered to launch oncogenic actions [138]. 

Since *G. vaginalis* is relatively highly abundant in the adolescent vagina and susceptibility to HPV infection increased during adolescence, it was suggested that these bacteria may be involved in the greater vulnerability observed [56,139]. Reduced protection provided by *L. iners* could be associated with the fact that it rarely produces antibacterial and antiviral H_2_O_2_ [140]. Moreover, menopause appears to contribute to HPV infection due to the decreased proportion of *Lactobacillus* spp. and higher microbiota diversity. In turn, based on the results of a longitudinal study of a cohort of 32 sexually active, premenopausal women, Brotman et al. [35] suggested that the predominance of *L. gasseri* in CST II could increase the rate of clearance of acute HPV infection. Clearance was defined as the transition from an HPV-positive state to a negative state. Thus, it appears that an abundance of *L. gasseri* may help preserve cervical health.

Tumour development is associated with many mechanisms, including the stronger and more cytoplasmic expression of TLR 2, 4, and 5 [141]. The proliferation of tumour cells and the development of cervical cancer were found to be stimulated by inflammatory cytokines such as IL-6, IL-8, and IL-1β [142,143]. Moreover, the aberrant activation of some signalling pathways triggers the occurrence of cervical cancer; for example, the activation of the Janus kinase (JAK)/STAT pathway contributes to immune escape. STAT3 enhances the expression of inhibitory cytokines involved in the regulation of immune homeostasis (TGF-β, IL-6, and IL-10), stimulates the aggregation of regulatory T cells, and hampers the maturation of dendritic cells, thus providing an immunosuppressive microenvironment for tumour development [144]. The expression of transcription factors may also be increased by HPV-related oncoproteins (E6/E7) [145]. The E6/E7 oncoprotein-induced dysregulation of NF-κB was found to stimulate aberrant cell proliferation and differentiation, inflammatory response, immune escape, angiogenesis, tissue infiltration, and metastasis [146]. Some studies pointed to oxidative and nitrifying stress as factors responsible for inflammation-induced tumours triggered by microorganisms [118,147]. Oxidative stress was found to constrain immune cell functioning. Nitrifying stress is associated with higher production of biogenic amines and nitrosamine and greater pathogen resistance to host defence systems [118]. Moreover, biogenic amines may facilitate the formation of bacterial biofilms. The abundance of *Lactobacillus* has been demonstrated to prevent the colonisation of amine-producing bacteria as well as to exert a cytotoxic effect on cervical cancer cells [45]. 

Zhang et al. [148] demonstrated that HPV16 E7 upregulated miR-27b to enhance the proliferation and invasion of cervical cancer. Moreover, hrHPV oncoproteins (E6/E7) stimulate the programmed cell death-1/programmed cell death-ligand 1 (PD-1/PD-L1) axis, thus leading to increased cancer progression. The checkpoint blockades that target PD-1/PD-L1 pathways have been found to hamper cancer development and improve survival, even in metastatic cervical cancer [149]. Summarized results of clinical studies demonstrating the impact of dysbiosis and HPV infection on cervical cancer development are presented in Table 1. 

Clinical studies have suggested that the majority of microbially driven carcinogenesis results from modifications of the microbiome rather than actions of a single pathogen [150]. Studies have shown the significance of dysbiosis in cancer types throughout the body. For example, the transfer of fecal material from patients with colorectal cancer into germ-free mice induced the hypermethylation of some genes in murine colonic mucosa. These alterations corresponded to those specific for the development of malignancy [151]. Microbial dysbiosis may trigger the tumourigenesis of various organs occupied by microorganisms, including the skin, lungs, and oral cavity [152]. 

## 6. Endometrial Cancer

Endometrial cancer (EC) is the fifth most frequent cancer in women, especially in developed or high-income countries [153,154,155]. This predominantly postmenopausal tumour originates in the endometrium in the inner epithelial lining of the uterus [156]. The endometrioid type of EC (especially endometrioid adenocarcinoma with oestrogen dependence) is the most frequent form that occurs in approximately 80% of cases, while non-endometrioid types (including, i.a., clear-cell EC, serous EC, carcinosarcoma, and other types) are much rarer [157]. The causative factors for this disease are not completely understood. Studies indicate that only 20% of endometrial cancer cases can be explained by genetics, including aerobic glycolysis impairment and the presence of microsatellite instabilities [158,159]. Moreover, environmental factors such as diabetes, obesity, inflammation, menopausal status, and gonadal hormones have been suggested to be involved in EC development [156]. Growing evidence suggests that microbiota present in the uterus can modify this organ functions in health and disease [65,160]. Indeed, the disruption of the “healthy” composition of uterine microbiota was found to be associated with infertility, endometritis, endometriosis, endometrial polyps, dysfunctional menstrual bleeding, and endometrial cancer [161,162,163]. Li et al. [164] suggested that the decreased diversity of the endometrial microbiome was associated with greater severity of this disease. The decreased α diversity of the microbiome was found to be associated with EC development [156]. Li et al. [164] demonstrated the positive correlation between higher abundance of endometrial *Prevotella* and increased serum D-dimer and fibrin degradation products. This finding may suggest high tumour burden. Another study found that the abundance of genera *Micrococcus* correlated with endometrial interleukin 6 and interleukin 17 messenger RNA levels, indicating the involvement of microbiota-inflammation crosstalk in EC development [114]. Inflammation appears to be a vital factor promoting the development of endometrial cancer [165]. It has been suggested that infiltrating inflammatory cells and local tissue are involved in cancer development [150]. Pelvic inflammation was suggested to accelerate the development of endometrial cancer [166]. The results of a nationwide, retrospective cohort study confirmed the role of pelvic inflammatory disease in the development of endometrial cancer [166]. Inflammation is involved in the endometrium remodeling cycle, and the released cytokines affect and alter endometrial mucosa [167]. Chronic inflammation-related mechanisms of elevated cancer risk may involve the promotion of free radicals formation leading to DNA damage, cell proliferation, and angiogenesis [168]. Available data indicate that microbiome may participate in the first stage of inflammation, triggering immunopathological changes, which finally lead to the development of cancer [169,170]. It appears that uterine microbiota can promote endometrial cancer development via the regulation of transcription factors and other epigenetic and genomic modifications, thus affecting the genomic stability of the uterine epithelium. Such modifications can hinder apoptosis and promote proliferation. Some microbiota are also capable of releasing genotoxins, damaging the host’s DNA and triggering cell carcinogenesis. Another possible mechanisms behind the relationship between disturbed uterine microbiota and endometrial cancer involves the production of bacterial toxins with tumour-promoting metabolites, which results in chronic bacterial inflammation and cytokine release by host cells [150]. The release of pro-inflammatory cytokines and antimicrobial peptides stimulate the development of the inflammatory response.

Endometrial cancer is proliferative disorder associated with hormonal dysfunction, including elevated levels of oestrogens and imbalances between progesterone and oestrogen production [171]. Such states favour uncontrolled profiling and hypertrophy and the subsequent development of endometrial cancer [172,173]. Alterations in microbiota composition can potentially result in the conversion of steroid molecules to potent androgens, thus leading to the formation of androgens and 11-oxyandrogens in EC patients [156]. In turn, Chen et al. [167] not only demonstrated that the abundance of 17 bacterial species differed between normal endometrium and EC but also that activated endometrial bacteria were engaged in EC metabolic processes (related to N-acetyl-β-glucosaminyl and 6-sulfo-sialyl Lewis x epitope) and tumour migration. Figure 2 presents mechanisms involved in the development of endometrial/cervical cancer.

Walther-António et al. [163] carried out a high-throughput comparative analysis of the microbiome present in the reproductive tract of females with benign uterine conditions, endometrial hyperplasia, and endometrial cancer. They observed a microbiome correlation between assessed organs (vagina/cervix, uterus, fallopian tubes, and ovaries). *Prevotella* and *Lactobacillus* were the dominant species inhabiting the vagina and cervix, while *Shigella* and *Barnesiella* were the most abundant in the uterus. In their study, the microbiome’s composition enabled a differentiation between benign uterine conditions and endometrial hyperplasia. This finding may imply that the microbiome can play a role in the early phases of cellular transformation. Since no significant differences were observed between the group of patients with endometrial cancer and hyperplasia or endometrial cancer and benign states, the authors suggested that, after transient disturbances in the microorganism profile, the microbiome reaches a new equilibrium [163]. Moreover, they revealed that the presence of *A. vaginae* and *Porphyromonas* sp. within the gynaecologic tract accompanied by low pH (>4.5) increased the risk of endometrial cancer. Finding these two bacteria in the uterus of females with hyperplasia, despite their absence in the lower tract, supports their role in the early stages of the disease. Moreover, other studies provide evidence for the involvement of *A. vaginae* in bacterial vaginosis, intrauterine, and other invasive infections of the female genital tract [174,175,176]. *A. vaginae*, which causes bacterial vaginosis, was suggested to elicit a prolonged inflammatory state that ultimately led to local immune dysregulation as well as the facilitation of intracellular infection by the *Porphyromonas* species. In turn, Walsh et al. [177] revealed that the presence of *Porphyromonas somerae* was highly predictive of concomitant uterine cancer. Another study demonstrated the abundance of *Micrococcus* sp. in the endometrial cancer group compared to benign uterine lesions (BUL) group, which was enriched in *Pseudoriibacter*, *Eubacterium*, *Rhodobacter*, *Vogesella*, *Bilophila*, *Rheinheimera*, and *Megamonas* [178].

## 7. Treatment

### 7.1. Vaccines

HPV type 16 (HPV-16) appears to be the most widespread form causing invasive cervical cancer [179,180,181]. Currently, most vaccines for preventing HPV have been developed based on a virus-like particle (VLP) derived from HPV L1. These vaccines provide protection against HPV infection; however, they are not effective in patients who have already been infected [182]. Therefore, there is a need for effective HPV vaccines that would promote immunogenicity against HPV oncoproteins. Various types of therapeutic vaccines have been developed, but the majority of them are based on the delivery of E6/E7 oncogenes via intramuscular or subcutaneous routes to trigger systemic immune response [183]. It has been observed that subcutaneous and intramuscular vaccines can augment systemic cellular immunity but not local mucosal immunity [184,185]. The effectiveness of lactic acid bacteria (LAB)-based vaccines also increases with the switch from injections to mucosal immunisation (intranasal, intravaginal, and oral) [186]. Finally, the magnitude of mucosal immune response depends on the number of viable colonies of LAB-expressing E6/E7 antigens [185]. Because genital mucosa is the key site for the entry of HPV-16, mucosally administered vaccines are being developed. According to current knowledge, the use of bacterial vaccines appears to be the most optimal option for the delivery of vaccine antigens to mucosal surfaces. However, in paediatric, elderly, and immunosuppressed patients, the delivery of live-attenuated bacterial pathogens may pose a risk [187]. 

LAB are gaining interest as live delivery vehicles. LAB has attracted attention as a potential component of HPV vaccines since it can be used to deliver antigens, and they upregulate the expression of IL-12 and IL-10, thus activating immature human bone marrow dendritic cells [188,189]. The NIsin-Controlled gene Expression (NICE) system in *Lactococcus lactis* appears to be the best choice currently available for the delivery of antigens at mucosal surfaces [179].

It has been demonstrated that these types of vaccines trigger strong humoral and mucosal immune responses against E6 and E7 oncogenic proteins [190,191]. Studies have indicated that exogenous target proteins (e.g., HPV-related protein) can easily attach to *Lactobacillus* S-layer signal peptides, thus enabling the development of a recombinant protein vaccine exerting antitumour effects [87]. The application of non-pathogenic and non-invasive *Lactococcus* spp., which are modified to deliver antigens of interest to mucosal surfaces, was suggested to induce beneficial therapeutic effects [192,193,194]. Such vaccines have been found to trigger both local and systemic immune responses; however, the stimulation within one mucosal site usually triggers a more pronounced response at that site than in distal mucosal sites [179]. Currently, there are several mucosal vaccines containing recombinant LAB targeting HPV-16 L1, L2, E2, E6, and E7 antigens [179]. Intranasal immunisation with live lactococci (expressing E7 antigen- and IL-12-triggered systemic and mucosal immune responses) protected mice against HPV-16-induced tumours [195]. Moreover, the oral use of *Lactobacillus* has been found to be beneficial and rarely causes side effects [87]. The results of an animal study demonstrated that the oral immunisation of mice administered with *L. lactis* harbouring HPV-16 L1 antigens was associated with the appearance of high levels of mucosal IgA antibodies [196]. *Lactobacillus casei* was suggested to be capable of the synthesis of recombinant L1 protein, which self-assembled into an intracellular virus-like particle (VLP) [197]. Another study provided evidence for the efficacy of triggering systemic and mucosal immune responses through a mixture of various forms of HPV-16 L1 protein produced by *L. lactis* [198]. Moreover, the N-terminal region of the L2 minor capsid protein of HPV-16 has been shown to have immune-boosting properties [199]. The oral immunisation with *L. casei* harbouring an anchored form of HPV-16 L2 protein was reported to trigger both L2-specific serum IgG as well as mucosal IgA antibodies [200]. LAB-based HPV vaccines have been found to exert an antitumour impact on HPV E6/E7-related neoplastic lesions in preclinical trials [179]. The triggering of strong mucosal immune responses within the cervix and the gastrointestinal tract by oral vaccination with recombinant LAB vaccines requires the stimulation of the galactose-1-phosphate uridylyltransferase gene (GALT) and integrin α4β7+ memory/effector cells. Such effects were observed following the consumption of *L. casei* with the HPV-16 E7 antigen [184]. Moreover, the oral administration of *L. lactis*-producing HPV-16 E6/E7 oncoproteins was associated with a higher amount of E6- and E7-specific IL-2- and IFN-γ-positive CD4+ and CD8+T cells in vaginal lymphocytes and intestinal mucosal lymphocytes, as well as a markedly enhanced immune response to major histocompatibility complex protein I (MHCI) (E6/7-specific CD8+ T cell) and II (MHCII) (E6/7-specific CD4+ T helper) epitopes [179,201,202]. In mice administered with *L. casei*-PgsAE6/E7, the impact on the immune system translated into a decreased tumour size and higher survival rates [203]. Even in mice vaccinated with a lethal dose of the tumour cell line TC-1, the administration of recombinant *L. lactis* resulted in antitumour protections and greater survival compared to control animals [201,202]. However, it has been revealed that a single immunisation with *L. lactis* may not be sufficient to elicit appropriate amounts of antigen-specific antibodies [196]. A double-blind, randomised, placebo-controlled phase I clinical trial enrolling healthy Iranian females demonstrated that an oral vaccine containing recombinant *L. lactis*-expressing codon-optimised HPV-16 E7 oncogene was associated with the production of HPV-16 specific serum-IgG and vaginal IgA antibodies, as well as cytotoxic T-lymphocyte responses in vaginal discharge and peripheral blood mononuclear cells [190,204]. Moreover, the phase I/IIa clinical trial comprising patients with CIN grade 3 (CIN3) indicated improved E7-specific cell-mediated immune responses in cervical lymphocytes to an *L. casei* vaccine containing a modified HPV-16 E7 antigen [191].

BioLeaders Corporation (South Korea) developed the BLS-M07 oral vaccine containing HPV-16 E7 antigen on the surface of *L. casei* for the treatment of CIN [179]. A clinical trial assessing its safety and efficacy in patients with CIN3 demonstrated that its use safely enhanced the production of serum HPV16 E7-specific antibody and, subsequently, improved humoral immunity [205].

However, the results of some studies indicated that mucosal and systemic immune responses may be affected by the antigen site in bacterial vectors [179,196]. In one study, the mucosal immune response was observed only in the case of intracellular production of HPV-16 L1 in *L. lactis* MG1363 [196]. In turn, Bermudez-Humaran et al. [206] reported a greater effect in cases of the extracellular expression of HPV-16 E7 in *L. lactis*. Several studies demonstrated that not only extracellular but also the cell-wall-anchored expression of a recombinant antigen greatly modulated systemic and vaginal immune responses [207,208,209]. The latter form of recombinant E6 and/or E7 protein was suggested to be associated with increased immune responses [201]. 

Poly-gamma-glutamic acid (γ-PGA) can be administered to enhance the antitumour effects of the oral *L. casei*-E7-based vaccine against cervical cancer [210]. Greater tumour suppression was also observed following intranasal pre-vaccination with recombinant *L. lactis*-expressing E7 in addition to adenovirus-expressing calreticulin-E7 (Ad-CRT-E7) in comparison to the use of the vaccine alone [211]. *Lactobacillus* is generally considered to be safe since it does not produce any toxic substances [212]. The results of animal studies and clinical trials confirm that the administration of recombinant LAB does not cause significant side effects [213,214].

The results of above-mentioned studies and trials are summarized in Table 2.

### 7.2. Probiotics and Prebiotics

Probiotics are defined as living microorganisms that are beneficial to the host organism [215,216]. They can be contained in conventional food, dietary supplements, infant formula, etc. [217]. Probiotics have been demonstrated to affect various biological processes associated with tumourigenesis such as inflammation, oxidative stress, apoptosis, proliferation, and metastasis [218,219,220]. Their utility has been suggested in the prevention and treatment of some diseases. Both probiotics and prebiotics (non-digestible food products that stimulate the growth of beneficial microorganisms in the intestines) exert beneficial properties, including anti-pathogenic, anti-inflammatory, antidiabetic, and immunostimulatory properties [221,222,223]. However, not all microbiota have been found to be beneficial, since some microorganisms may be involved in carcinogenesis [216]. *Lactobacillus* bacteria and their products can hinder cervical cancer proliferation with respect to their impacts on immunological mechanisms and cancer-related pathways. *Lactobacillus* potentiates the antitumour effects of macrophages, T cells, dendritic cells (DCs), and NK cells [224,225]. They also stimulate innate immune responses and can selectively accumulate within hypoxic zones of solid cancers [226,227]. *Lactobacillus* supernatants, *L. crispatus*, *L. jensenii*, and *L. gasseri* have been demonstrated to constrain the proliferation of CaSki cells [228]. This study showed a marked increase in the number of S phases as well as a reduction in G2/M phase cells following cell incubation with *Lactobacillus* supernatants. Moreover, *Lactobacillus* supernatants diminished the expression of cyclin A, CDK2, and E6/E7 HPV oncogenes that are necessary for the transition into malignancy [228,229]. Nami et al. [230] revealed probiotic and anticancer properties of *L. plantarum* species isolated from vaginal secretions of adolescent and young adult women. This strain displayed antibiotic susceptibility and antimicrobial actions against some pathogenic bacteria. Moreover, it showed outstanding anticancer activity in cases of human cancer cell lines; however, no visible cytotoxic effects on normal human umbilical vein endothelial cells (HUVEC) were observed [230]. Another study indicated similar antimicrobial and anticancer properties of *Lactobacillus* strains (*L. casei* SR1, *L. casei* SR2, and *Lactobacillus paracasei* SR4) isolated from human milk [231]. These bacteria promoted the upregulation of apoptotic genes (caspase3, caspase8, caspase9, BAD, and BAX) as well as the down-regulation of BCL-2. Moreover, *L. gasseri* strains (G10 and H15) found in the human vagina hindered the proliferation of HeLa cells [232]. These strains, through a decrease in TNF-α and an increase in IL-10, exerted an anti-inflammatory effect on cervical cancer. *L. rhamnosus* and *L. crispatus* can also diminish the expression of MMP2, MMP9, and caspase 9, thus hampering metastasis [233]. *L. crispatus* was found to limit E6/E7 expression at the level of miRNA, while *L. gasseri* acted only on the E6 gene [234]. Probiotic bacteria can also boost the effect of antitumour treatment, e.g., cisplatin therapy in patients with advanced cervical cancer [235]. Improved responses to cisplatin treatments were associated with the upregulation of interferon γ (IFN-γ), perforin 1 (PRF1), and granzyme B (GZMB), expressed by cytotoxic T lymphocytes and NK after the administration of *Lactobacillus* [236]. Hummelen et al. [237] demonstrated that oral administration of both *L. rhamnosus* GR-1 and *L. reuteri* RC-14 protected against bacterial vaginosis or cured it due to an upsurge in the amount of dominant *Lactobacillus* in vaginal microbiota. The mechanism via which these extraneous species can alter community structure may involve the action of bacteriocins produced by both bacteria (*Gasseri* or *Lactocin*) [238]. In addition to living bacteria, a biological response modifier (LC9018) isolated from heat-killed *L. casei* YTT9018 was also found to improve the effects of radiation therapy used in a group of patients with carcinoma of the uterine cervix (Stage IIB or III). This combination therapy was associated with a greater reduction in tumour size compared to radiation therapy alone and also appeared to protect patients from leukopenia during radiotherapy [239]. Moreover, patients receiving LC9018 displayed higher survival and longer relapse-free interval compared to those treated with radiation alone. 

The knowledge of exact mechanisms associated with antitumour actions of probiotic bacteria is still limited and requires further research. The dysregulation of numerous miRNA (including miR-21, miR-29a, miR-9, miR-10a, miR-16, miR-20b, miR-106, miR-375, miR-125, and miR-34a) have been reported in the course of cervical cancer [240]. Growing evidence suggests that *Lactobacillus* and other strains isolated from the vagina can positively affect the regulation of, e.g., TLR-4, miR-21, and miR200b, thus stimulating apoptosis [241]. 

## 8. Future Perspectives and Limitations

Despite advances in understanding the associations between microbiota dysregulation and carcinogenesis, further studies are required to unravel the underlying mechanisms and confirm the previous findings in prospective studies of a large population. Moreover, future research should also focus on strategies to manipulate vaginal microbiota to decrease cervical cancer risk. There is a need for prospective studies that would assess the incidence of cervical cancer after the administration of a *Lactobacillus* vaccine. Future research should also focus on the understanding of the molecular mechanism via which *Lactobacillus* constrains the proliferation of cells and cervical cancer. Since *Lactobacillus* is used as a carrier to express alternative antigens, it is plausible that they could be used to express antioncogenes or encapsulated anticarcinogens. 

The cross-sectional nature of most studies may limit the ability to identify a causal association between the composition of the vaginal microbiota, HPV infection, and CIN/cervical cancer. Moreover, cervical cancer develops for years (or even decades) from the initial acute HPV infection, which makes studies in this field more difficult. Furthermore, since the depletion of *Lactobacillus* spp. can be associated with various factors such as smoking and vaginal intercourse without barrier contraception, studies in this field should be carefully designed to ensure that observed disturbances in microbiota profiles were triggered by dysbiosis. Due to the possibility of cross-contamination, the collection of samples can also affect the final effects of the study. 

Future research should focus on interactions between microbiota and the host immune system and also consider HPV infection history. A deep understanding of the mechanisms involved in the interactions between vaginal microbiota and the host immune system may also provide an explanation for HPV persistence and subsequent neoplastic transformation. Confirmations are also required with respect to whether some strains of bacteria can exert protective/pathogenic effects in cases of HPV and cervical dysplasia. Finally, the therapeutic efficiency of probiotics/prebiotics in the treatment of high-grade CIN should be assessed.

## 9. Conclusions

Gut microbiota can affect the risk of developing some diseases. Therefore, the formation of infant gut microbiota is of high importance. The development of early gut microbiota can be regulated by numerous factors, including the method of child delivery, host genetics, gestational age, and specific dietary compounds present in human milk. Microbiota inhabiting the vagina seem to modulate the acquisition and persistence of HPV, affecting the risk of CIN development and progression. Bacteria residing in our body, especially *Bifidobacteria,* were found to be specifically adapted to using glycan components available in the human body to produce lactic acid, which protects humans against colonisation by pathogens. This indicates the symbiosis between the host and microbiota. Numerous studies demonstrated that alterations in microbial diversity and disturbances in microbiome composition may be associated with the development of various diseases. However, the exact mechanisms underlying this phenomenon are not fully understood. The identification of “healthy” microbiota compositions may offer an opportunity to develop novel therapeutic agents (probiotics) that could help prevent HPV infection, stimulate its clearance in infected women, and significantly reduce the risk of cervical dysplasia. Studies have already indicated the potential of probiotics in either the prevention or the treatment of cervical cancer since they were demonstrated to promote apoptosis, decrease inflammation, hinder proliferation, and suppress metastasis. Probiotics appear to exert more pronounced effects if combined with anti-infective drugs. Future research should focus on microbiota-mediated immune and physiological responses related to the development of diseases, including cervical cancer. The identification of microbial biomarkers enabling the prediction of disease development and early implementation of appropriate measures to prevent disease progression is also necessary. 

## Figures and Tables

**Figure 1 cancers-14-04909-f001:**
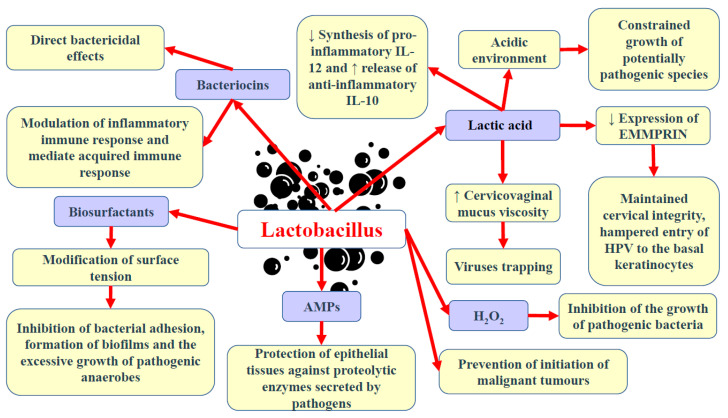
Basic beneficial mechanisms of Lactobacillus in the female genital tract. Abbreviations: ↓—decrease; ↑—increase; AMPs—antimicrobial peptides; H_2_O_2_—hydrogen peroxide; EMMPRIN—extracellular matrix metalloproteinase inducer; IL-10 and -12—interleukin-10 and -12.

**Figure 2 cancers-14-04909-f002:**
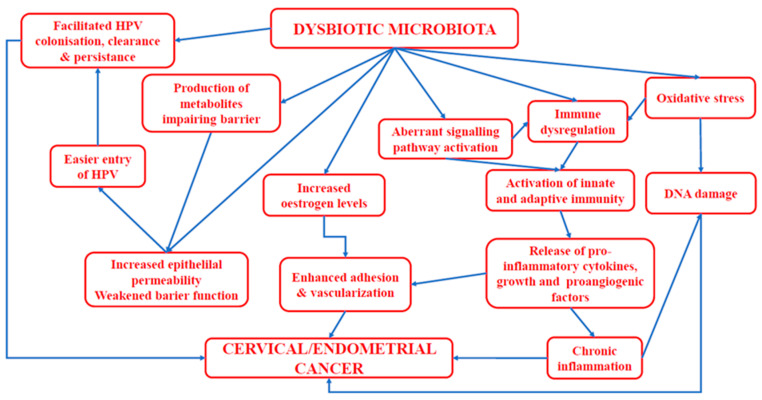
Mechanisms involved in the onset of endometrial/cervical cancer.

**Table 1 cancers-14-04909-t001:** Results of studies demonstrating the impact of dysbiosis and HPV infection on cervical cancer development.

Type of Study	Studied Population	Main Results	Ref
Open, single-site study	32 women aged 38–55 years with established cervical cancer (FIGO I stage)	- Disturbances of vaginal microbiota occurred in 71% of patients with FIGO I stage cervical cancer.	[120]
Oriented observational, prospective, cohort study	85 women with CIN2/CIN3 diagnosis, candidates for LEEP	- CIN2: microbiome dominated by *Lactobacillus* spp., but a high presence of anaerobic Gram-negative BV-associated bacteria (especially *A. vaginae, G. vaginalis,* and *Ureaplasma parvum*) and less widespread microbes, including *Candida albicans, Finegoldia magna, Peptoniphilus asaccharolyticus, P. anaerobius, Prevotella bivia,* and *Streptococci*, was observed. - CIN3: reduction in lactobacilli, except for *L. iners*, and high prevalence of *A. vaginae, G. vaginalis,* and *U. parvum*, as well as *Aerococcus christensenii, Anaerococcus prevotii, Leptotrichia amnionii, M. hominis, Parvimonas micra, Peptoniphilus asaccharolyticus, Porphyromonas asaccharolitica, P. bivia, Prevotella buccalis,* and *S. sanguinegens.* - High concentration of pro-inflammatory cytokines in the vaginal environment of CIN patients, including IL1α, IL1β, IL6, IL8, and TNFα, confirming that BV-like vaginal microbiomes are associated with increased local inflammation. - Surgical removal of hrHPV-related CIN lesions per se triggered microbiome remodulation.	[123]
In vitro study of cervical cancer cell lines C33a (HPV-), SiHa and CaSki (HPV16+), and HeLa (HPV18+) cells	120 fresh cervical tissue biopsies (70 malignant, 30 premalignant, and 20 normal (control) cervical tissues)	- Aberrantly expressed and constitutively active STAT3 was found both in cervical cancer cell lines and in cervical precancer and cancer lesions. - Increased expression of STAT3 was regulated at transcription level. - Concurrent raise in phosphorylation at Tyr705 and Ser727 responsible for the regulation of STAT3 dimerization, nuclear transport, and DNA-binding and transactivation. Dually phosphorylated STAT3 present in cervical precancer and cancer lesions was found to localise to the nuclei and possessed a functional DNA-binding activity. - STAT3 expression and activation correlated well with HPV16 positivity in cervical precancer and cancer lesions. - Activation of STAT3 in cervical cancer cases increased along with disease severity.	[129]
Prospective study	23 HPV-positive and 45 HPV-negative women who participated in the Healthy Twin Study	- The percentage of *Lactobacillus* spp. was considerably decreased in the HPV-infected group. - Higher diversity of vaginal microbiota of the HPV-positive group compared with the HPV-negative group. - HPV infection strongly correlated with the abundance of various vaginal microbiota species, e.g., *Prevotella, Sneathia, Dialister,* and *Bacillus*. - *Sneathia* spp. was a microbiological marker of high-risk HPV infection. - 17% of HPV-positive premenopausal women had CIN (a potential precursor of cervical cancer).	[131]
Prospective cohort study	169 women: healthy (n = 20), low-grade squamous intraepithelial lesion (LSIL) (n = 52), high-grade squamous intraepithelial lesion (HSIL) (n = 92), and ICC (n = 5).	- 2-fold increase in the rate of a CST IV vaginal microbiome in women with LSIL; 3-fold increase in women with HSIL; 4-fold increase in women with invasive cancer compared to controls. - Presence of HSIL markers *P. anaerobius* and *A. tetradius*. - Presence and predominance of specific vaginal microbiome CSTs may be involved in the pathogenesis of CIN and cervical cancer.	[133]
A cross-sectional study	32 cases: non-cervical lesions (NCL: n = 10 HPV-negative; n = 10 HPV-positive), SILs (n = 4 HPV-positive), and CC (n = 8 HPV-positive)	- Cervical microbiome is notably different in all stages of the natural history of cervical cancer. - Higher median cervical levels of IL-4 and TGF-β1 mRNA in CST VIII, dominated by *Fusobacterium* spp. - *Sneathia* spp., *Megasphaera elsdenii,* and *S. satelles* were most representative in the SIL cases.	[7]
A systematic review and network meta-analysis	Analysis of 11 included studies	- Vaginal microbiota dominated by non-lactobacilli species or *L. iners* were associated with 3–5-times higher odds of any prevalent HPV and 2–3-times higher for hrHPV and dysplasia/cervical cancer compared with *L. crispatus*.	[135]
Prospective study	50 cervicovaginal swab specimens obtained from women aged 20 to 50 (40 positive for hrHPV and 10 negative for hrHPV)	- Abundance of *Lactobacillus* species was decreased in women with cervical disease; the amount of *L. crispatus* was significantly reduced in women with CIN and cervical cancer. - Markedly increased abundance in anaerobic bacteria: *G. vaginalis, P. anaerobius*, and *Porphyromonas uenonis* in women with CIN and cervical cancer. - Presence of *G. vaginalis* is associated with a high risk for developing CIN 2 or 3 and cervical cancer.	[137]
In vitro study	Clinical samples obtained from six HPV16-positive cervical cancer patients, HPV16-positive human cervical carcinoma cell lines CaSki and SiHa, and HPV-negative cervical cancer cell line C33A	- Increased miR-27b expression levels in cervical cancer tissues compared to adjacent normal tissues. - miR-27b-enhanced proliferation and invasion of cervical cancer cell lines, confirming that miR-27b serves as an oncogene in cervical cancer. - Inhibition of PPARγ-promoted proliferation and invasion of cervical cancer cells, both antitumour roles of PPARγ in cervical cancer. - miR-27b was positively regulated by HPV16 E7. - miR-27b inhibited the expression of PPARγ. - Overexpression of HPV16 E7 suppressed the expression of PPARγ depending on the existence of miR-27b; HPV16 E7 is able to repress the expression of PPARγ through the stimulation of miR-27b.	[148]

**Table 2 cancers-14-04909-t002:** Lactic acid bacteria (LAB)-based vaccine studies.

Vaccines				
Animal Studies				
Studied Agent	Route of Administration	Type of Study	Observed Effects	Ref
Recombinant *Lactobacillus casei* expressing HPV16 E7 (LacE7)	Mucosal (oral)	Animal study	- Elicit E7-specific IFN gamma-producing cells (T cells with E7-type 1 immune responses) - Greater induction of T cells compared to subcutaneous or intramuscular antigen delivery. - Trigger mucosal cytotoxic cellular immune responses	[185]
*L. lactis* MG1363 was transformed with two types of HPV16 L1-encoding plasmids for intracellular expression or secretion.	Oral	Animal study	- Serum IgG responses after immunizations with *L. lactis* secreting HPV16 L1. - Vaginal IgA immune responses after oral immunization with L. lactis expressing HPV16 L1, but secreting HPV - HPV16 L1-specific mucosal immune responses affected by immunization frequency.	[197]
N-terminal L2 polypeptides comprising residues 11 to 200 derived from HPV16 produced in bacteria (HPV16 L2 11–200)	Vaccination	Animal study	- Effective protection of rabbits against cutaneous and mucosal challenge with CRPV and ROPV - Generation of broadly cross-neutralizing serum antibody - potential of L2 as a second-generation preventive HPV vaccine antigen.	[200]
A partial HPV-16 L2 protein (N-terminal 1–224 amino acid) on the surface of *L. casei*.	Mucosal (oral)	Animal study	- Production of L2-specific serum IgG and vaginal IgG and IgA in Balb/c mice - Trigger systemic and mucosal cross-neutralizing effects in mice	[201]
*L. lactis* NZ9000 expressing *human papillomavirus type 16* E7 antigen	Mucosal (oral)	Animal study	- Elicit the highest levels of E7-specific antibody and numbers of E7-specific CD4+ T helper and CD8+ T cell precursors. - Potent protective effects against challenge with the E7-expressing tumour cell line (TC-1) - pNZ8123-HPV16-optiE7 containing *L. lactis* showed strong therapeutic antitumour effects against established tumours in vivo. - Trigger humoral and cellular immune responses in mice	[202]
Recombinant strains of *L. lactis* NZ9000 expressing native and codon-optimized E6 protein (fused to the SPusp45 secretion signal)	Mucosal (oral)	Animal study	- Improved inhibitory effect on tumour growth, improved treatment effects on progression of tumour size, and improved survival rates in comparison with *L. lactis* having native E6 oncogene - Induce humoral and cellular immunity	[203]
HPV16 E7 antigen expressed on the surface of *L. casei*	Mucosal (oral) vaccine	Animal study	- Enhanced E7-specific serum IgG and mucosal IgA production. - Reduced tumour size and increased survival rate in E7-based mouse tumour model compared to versus mice receiving control (*L. casei*-PgsA) immunization.	[204]
HPV16 E7-expressing *L. casei* (*L. casei*-E7) combined with γ-PGA secreted by *Bacillus subtilis*	Mucosal (oral) vaccine	Animal study (TC-1 mouse model)	- Enhanced innate immune response including activation of dendritic cells - Significantly suppressed growth of TC-1 tumour cells and an increased survival rate compared to mice vaccinated with L. casei-E7 alone. - Markedly enhanced activation of natural killer (NK) cells, no impact on E7-specific cytolytic activity of CD8+ T lymphocytes.	[211]
Combination of adenovirus expressing calreticulin-E7 (Ad-CRT-E7) and *L. lactis* encoding HPV-16 E7 (Ll-E7) anchored to its surface	Intranasal preimmunization of Ll-E7, followed by a single Ad-CRT/E7 application	Animal study (mouse model)	- ∼80% of tumour suppression compared to controls. - 70% survival rate 300 days post-treatment (100% of controls died by 50 days). - Significant CD8+ cytotoxic T-lymphocytes infiltration in tumours of mice treated with Ll-E7+Ad-CRT/E7.	[212]
**Clinical Studies and Trials**				
Attenuated *L. casei* expressing modified full-length HPV16 E7 protein	Oral (during dose optimization studies (1, 2, 4, or 6 capsules/day) at weeks 1, 2, 4, and 8 (n = 10) or optimized vaccine formulation (n = 7)	Patients with HPV16-associated CIN3	- Most patients (70%) receiving the optimized dose experienced a pathological down-grade to CIN2 at week 9 of treatment - E7-specific mucosal immunity was elicited in the uterine cervical lesions.	[192]
NZ8123-HPV16-optiE7 vaccine involving recombinant *L. lactis* expressing the codon-optimized human papillomavirus (HPV)-16 E7 oncogene	Oral vaccine or placebo	A dose-escalation, randomized, double-blind, placebo-controlled phase I clinical trial was performed in healthy Iranian volunteer women	- Vaccination was well tolerated, and no serious adverse effects were reported - Dose-dependent response to NZ8123-HPV16-optiE7 vaccine following oral administration - Safety and immunogenicity profile achieved in this study encourages further phase II trials with the 5 × 109 CFU/mL dose vaccine	[205].
BLS-M07 (HPV 16 E7 antigen expressed on the surface of *L. casei*)	Oral administration Phase 1: 5 times a week, on weeks 1, 2, 4, and 8 with dosages of 500 mg, 1000 mg, and 1500 mg Phase 2a: 1000 mg dose.	A phase 1/2a, dose-escalation, safety, and preliminary efficacy study performed in patients with CIN 3	- No dose limiting toxicity. - No grade 3 or 4 treatment-related adverse events or deaths - Improved RCI grading (16 weeks after treatment) - Increased serum HPV16 E7 specific antibody production.	[206]

## Data Availability

Not applicable.

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
