# Peer review of "Does Lactobacillus Exert a Protective Effect on the Development of Cervical and Endometrial Cancer in Women?"

_cancers, 2022, doi:10.3390/cancers14194909_

Round 1
Reviewer 1 Report
The review by Fraszczak et al. addresses the role of microbiota in cancer risk and discusses up-to-date methods for combating cervical and endometrial malignancies. This is an excellent review that will pique the readers' interest. However, a few suggestions will increase the quality of the manuscript.
1. The author should go into more detail on endometrial cancer as well as elaborate more on how pelvic inflammation might hasten the growth of endometrial cancer.
3. There are lots of grammatical and spelling mistakes in the whole review. Minor changes to the manuscript's writing will also help to reinforce the paper.
Author Response
- We provided additional information on endometrial cancer and the pelvic inflammation as a factor accelerating the growth of endometrial cancer.
- The manuscript was proofread to correct all the grammatical and spelling mistakes.
Reviewer 2 Report
This is a interesting review article written by Bartlomiej Barczynski and Colleague.
The review article is well written and elaborately explained about the Lactobacillus exert a protective effect on the development of cervicl and endometrial cancer in women.
I recommend this article for publication after minor revision.
Comments: Please change the figure in high pixel (The letters are not legible for reading).
There are some typos, please rectify it.
Author Response
- The resolution of the Figure 1 was improved.
- The manuscript was proofread to correct all the grammatical and spelling mistakes.
Reviewer 3 Report
I read the paper by Fraszczak et al. This review is intriguing and well organized. However minor revision are needed before pubblication.
First of all Figure 1 is difficult to understanding and low resolution. It should be improved.
Table 1 is not present in my version, so I cannot evaluate it .
Paragraph 6. In this section should be insert a figure that explain involvment of microbiota in endometrial cancer
Paragraph 7. A table should be insert
Author Response
- The resolution and graphical design of the Figure 1 was improved. The violet fields contain molecules secreted by bacteria, while the yellow ones – mechanisms of action.
- We would like to apologize for this inconvenience. The Table 1 was added.
- We provided the Fig.2 presenting the role of dysbiosis in endometrial cancer development.
- We prepared additional Table 2 summarizing the described studies involving vaccines based on L. casei.